# Multichannel Analysis of Surface Waves (MASW) for the internal characterisation of the Flüela rock glacier: overcoming the limitations of seismic refraction tomography

Ilaria Barone[1], Alexander Bast[2,3], Mirko Pavoni[1], Steven Javier Gaona Torres[1], Luca Peruzzo[1] and Jacopo Boaga[1]

[1] Department of Geosciences, University of Padova, Padua, Italy

[2] WSL Institute for Snow and Avalanche Research SLF, Permafrost Research Group, Davos Dorf, Switzerland

[3] Climate Change, Extremes and Natural Hazards in Alpine Regions Research Center CERC, Davos Dorf, Switzerland

*Correspondence to*: Ilaria Barone (ilaria.barone@unipd.it) and Mirko Pavoni (mirko.pavoni@unipd.it)

**Abstract.** A multi-method geophysical campaign was carried out to characterize the subsurface of the Flüela rock glacier in Grisons, Switzerland, using electrical resistivity tomography (ERT), seismic refraction tomography (SRT), and multichannel analysis of surface waves (MASW). Surface-wave analysis is not commonly applied in mountain permafrost environments, although it can be used on any dataset acquired for conventional SRT when low-frequency geophones are employed. Here, we show that the MASW method can be effectively used to highlight the presence of an ice-bearing layer, thereby overcoming potential limitations of conventional SRT in these environments, such as noisy first arrivals, attenuation effects, and velocity inversions at depth. Our results are corroborated by synthetic ERT and full-wave seismic modelling, which independently support our subsurface interpretation.

Keywords: multichannel analysis of surface waves (MASW); electrical resistivity tomography (ERT); seismic refraction tomography (SRT); mountain permafrost hydrology; rock glacier hydrology; ground ice content.

## 1 Introduction

The warming and degradation of European mountain permafrost (PERMOS, 2020; Noetzli et al., 2024) facilitates the formation and dynamics of alpine mass movements such as rock falls, landslides or debris flows (Arenson and Jakob, 2014; Kofler et al., 2021; Bast et al., 2024a; Jacquemart et al., 2024), and hence, may impact human safety and infrastructure (Arenson and Jakob, 2017; Duvillard et al., 2019). Consequently, in densely populated mountain regions such as the European Alps, there is a significant demand for reliable tools to map and characterise permafrost environments, accurately assess associated risks, and apply practical solutions for the construction and maintenance of durable infrastructure (e.g., Bommer et al., 2010).

Rock glaciers are common, widespread, often tongue-shaped debris landforms found in periglacial mountain environments containing ice, rocks, air and water (Kellerer-Pirklbauer et al., 2024; Haeberli et al., 2006;

RGIK, 2022). They form in the deposition zones of snow avalanches and rock fall (Kenner et al. 2019) and develop over centuries to millennia (Krainer et al., 2015; Haeberli et al., 1999) due to past or ongoing creep (RGIK, 2022), resulting from internal deformation within the ground ice and shearing at distinct horizons (Arenson et al., 2002; Cicoira et al., 2021). In the past two decades, the creep rate of rock glaciers has generally increased, and this is often linked to climate change (Kellerer-Pirkelbauer et al., 2024; PERMOS 2020, Hu et al. 2025).

Rock glaciers have primarily been studied from geomorphological, climatic, and kinematic perspectives, with less focus on their hydrological aspects (e.g., Bast et al., 2024b; Cicoira et al., 2019; Haeberli et al., 2006; Hu et al., 2025; Kellerer-Pirklbauer et al., 2024; Kenner et al., 2020), as also highlighted by recent reviews by Arenson et al. (2022) and Jones et al. (2019). This gap in understanding arises because of the complexity of the distribution of ice and water in rock glaciers. The relation between rock glacier kinematics and their hydrology is also complex, influenced by factors such as variable surface cover and groundwater flow, which affect infiltration rates, heat transfer and reaction times (Arenson et al., 2022). Nevertheless, understanding rock glacier hydrology is essential to comprehend rock glacier velocities, i.e. kinematics, and their potential impacts on alpine mass movements.

Water can exist within rock glaciers as seasonally frozen in the active layer, as perennially frozen ice in the permafrost body, or perennially unfrozen in liquid form in taliks. Permafrost primarily influences water flow paths by acting as a physical barrier that restricts movement (Arenson et al., 2022). Conceptual models (Giardino et al., 1992), alongside geochemical (Krainer and Mostler, 2002; Krainer et al., 2007) and geophysical studies (Pavoni et al., 2023a), suggest that a continuous ice-rich frozen layer functions as an aquiclude, separating supra-permafrost flow caused by snow and ice melt, as well as precipitation, from a deeper sub-permafrost flow (Jones et al. 2019). However, the stratigraphy and the bedrock under rock glaciers are often very heterogeneous over short distances, complicating the hydrology (Bast et al., 2024b; Boaga et al., 2020). The thermal state of the ground also plays a critical role, as liquid water can exist below 0°C due to factors such as water salinity, high clay content, or pressure (Arenson et al., 2022; Arenson and Sego, 2006; Bast et al., 2024b; Williams, 1964). This affects the unfrozen water content and hydraulic conductivity and may lead to intra-permafrost flow, confined water layers or water pockets. Furthermore, heat transport by flowing water can facilitate thawing in specific regions, for instance, leading to the development of taliks (Arenson et al., 2022).

Although boreholes provide the most accurate information on the internal structure of rock glaciers (Arenson et al., 2002) and allow the monitoring of subsurface properties through specialised sensors such as high-accuracy piezoresistive level probes with temperature sensors or inclinometers (Bast et al., 2024b; Phillips et al., 2023; Arenson et al., 2002), they only offer point data, they are expensive and are challenging to install in high mountain environments. Geophysical methods are, therefore, often used to achieve a more detailed characterisation of the subsurface and a spatial extent (e.g., Scott et al., 1990; Hauck and Kneisel, 2008).

Among the different geophysical techniques, electrical resistivity tomography (ERT) and seismic refraction tomography (SRT) methods are widely used to estimate the structure and internal composition of rock glaciers (Wagner et al., 2019; Pavoni et al., 2023b; Hauck et al., 2011; de Pasquale et al., 2022). Single-station passive seismic methods such as HVSR (Horizontal-to-Vertical Spectral Ratio) are also increasingly popular for permafrost characterization and monitoring (Kula et al. 2018), including in rock glacier environments (Guillemot et al. 2020, Guillemot et al. 2021, Colombero et al. 2025). Among the advantages of passive seismic methods there is the simplified logistics, which is counterbalanced by the point-station character of the measurement and

the lack of high frequencies, resulting in a reduced sensitivity in the very near surface. On the other hand, the multichannel analysis of surface waves (MASW; Park et al., 1999), commonly applied for civil engineering purposes (Park et al., 2018; Olafsdottir et al., 2024) and recently used in permafrost studies in Arctic regions (Glazer et al. 2020, Liu et al. 2022, Tourei et al. 2024), has rarely been applied in mountain permafrost environments (Guillemot et al., 2021; Kuehn et al., 2024). Nevertheless, a seismic shot gather acquired with low-frequency vertical geophones (e.g., with a 4.5 Hz natural frequency) not only records the first arrivals of direct and refracted P-waves but also Rayleigh waves, whose propagation is mainly sensitive to S-wave velocities (Vs). Thus, the application of the MASW method can potentially allow the retrieval of a Vs profile (Socco et al., 2010), complementing the SRT method, which typically focuses on P-wave velocities (Vp). The MASW method offers several advantages over the SRT technique: i) it can reveal velocity inversions in the subsurface, such as a lower velocity layer between two higher velocity layers, ii) the retrieved S-wave velocities are insensitive to the liquid phase present in the medium, and iii) it can provide quantitative information regarding the subsurface mechanical properties like the shear modulus and Young's modulus, for geotechnical characterisation (Park et al., 2007).

In this study, we applied the MASW method along a seismic line acquired next to an ERT line at the Flüela rock glacier, Grisons, Switzerland. ERT suggests the presence of an ice-bearing layer in the upper part of the rock glacier tongue, which disappears towards the front. The SRT analysis clearly detects the basal bedrock but surprisingly does not reveal the typical P-wave velocities of the ice-bearing layer. In fact, the SRT results indicate Vp values typical of liquid water, thereby masking the presence of the ice-bearing layer. In contrast, the Vs models obtained from the MASW results are in very good agreement with the ERT findings. We therefore hypothesise that the difficulties encountered in the SRT analysis in detecting the ice-bearing layer are due to the presence of a thin water-saturated sediment layer overlying the ice-bearing layer (supra-permafrost flow), which would inhibit P-wave propagation, as well as relatively high picking uncertainties. To support our hypothesis, we performed both full-wave seismic forward modelling, producing synthetic shot gathers for comparison of surface-wave dispersion and P-wave first-arrival times, and synthetic ERT modelling to evaluate the capacity of the adopted ERT array to resolve the thin water-saturated layer above the permafrost.

## 2 Study site and data acquisition

The lower lobe of the Flüelapass rock glacier complex (referred to here as the Flüela rock glacier; 46.746° N, 9.951° E) is located in the Eastern Swiss Alps, next to the Flüelapass road in the Community of Zernez, Grisons, at the top of the mountain pass (Figs. 1a and 1b). The active rock glacier, ranging from 2380-2500 m asl., is nourished by the surrounding steep rock walls, which are composed of amphibolite and paragneiss (Bast et al., 2025). The lower investigated tongue of the rock glacier (Fig. 1c) creeps downwards with surface velocities ranging between ~ 10 and ~ 30 cm/year (R. Kenner, SLF, personal communication, based on annual terrestrial laser scans, 2024). The surface material consists of rock debris and boulders of various sizes, along with smaller isolated patches of finer sediments and sparse vegetation (Figs. 1c and 1d).

A first study of the Flüela rock glacier by Haeberli (1975) applied refraction seismics to investigate the presence of ground ice. The seismic profiles obtained indicated permafrost at around 10 m depth in the rock glacier front and ice-rich ground below approximately 4 m towards the central lower area of the rock glacier. More recent

geophysical research by Boaga et al. (2024) and Bast et al. (2025) confirmed the presence of the ice-bearing layer.
Research on permafrost distribution and evolution at the Flüelapass primarily concentrated on a talus slope located
approximately 500 m west-northwest, where two boreholes were drilled in 2002 (Lerjen et al., 2003; Phillips et
al., 2009; Kenner et al., 2017). As for the lower tongue investigated here, no borehole information is available.

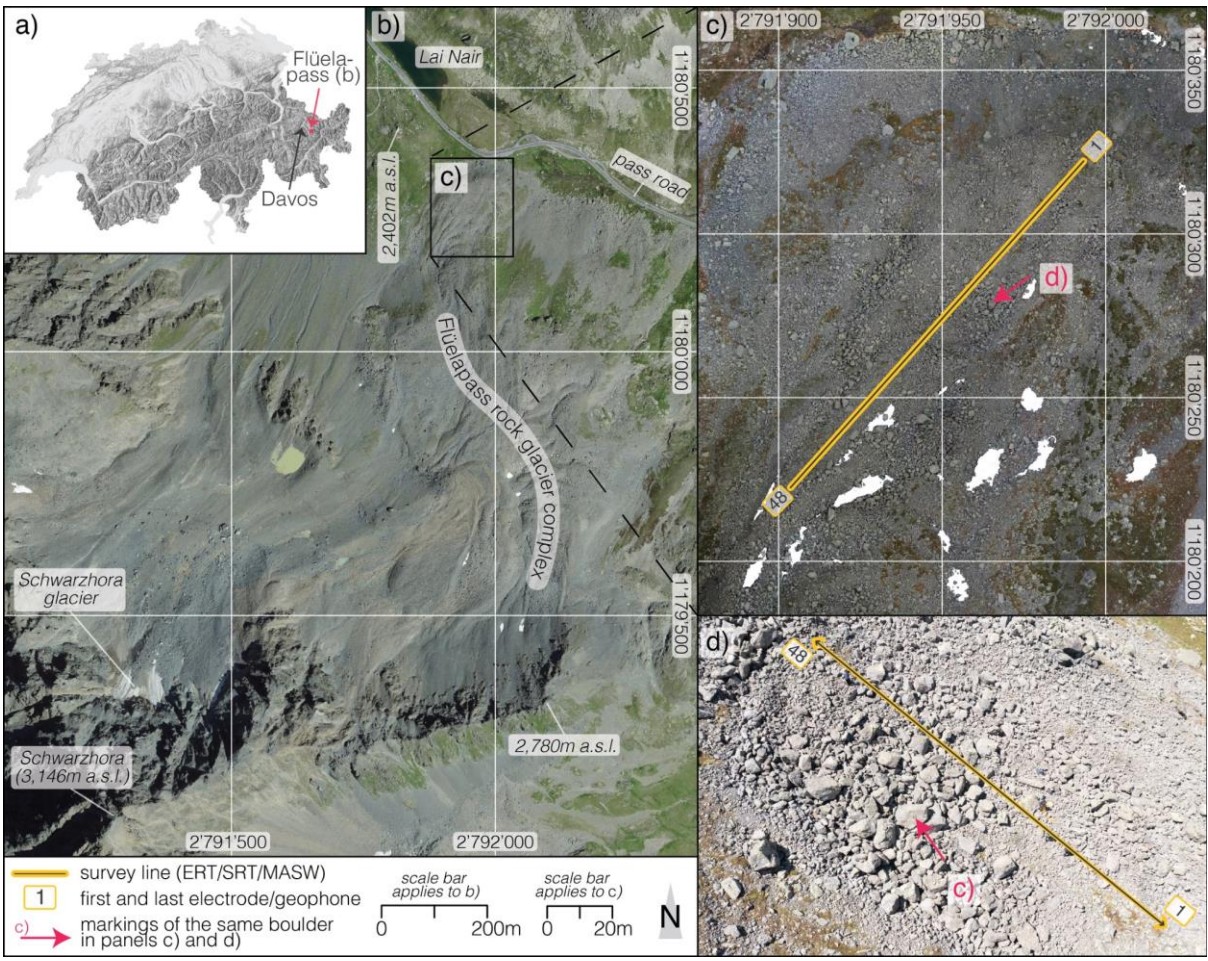


**Figure 1: (a) Location of the Flüela rock glacier complex in Switzerland. (b) Aerial image of the Flüelapass featuring**
**the Flüela rock glacier complex, with markers for orientation (summit, lake, road) and elevation points. (c) A zoomed-**
**in drone ortho mosaic (flight date 28 June 2023) of the investigated lower lobe of the rock glacier complex, highlighting**
**the survey line (electrical resistivity and seismic data, yellow-grey line). The red arrow indicates the same boulder as**
**in the oblique drone image (d). The drone image reveals the coarse and rough surface within the middle section of the**
**survey line (please note that the survey line extends further NW and SW). Basemaps in a) SwissAlti3D multidirectional**
**hillshade and b) SwissImage (flight year 2022) are provided by swisstopo ([https://map.geo.admin.ch](https://map.geo.admin.ch)). Note that the**
**legend and North arrow applies to all map sections (a-c).**
On 03 August 2024, we collected both electrical resistivity tomography (ERT) and seismic data on the rock
glacier. The measurements were collected along a line of approximately 133 m in the middle of the lobe (Figs. 1c
and 1d). For data collection, we used 48 electrodes for the ERT and 48 geophones for the seismics, with a spacing
of 3 m. We measured all electrode/geophone positions with a Stonex S800 GNSS instrument (Stonex, Paderno
Dugnano, Italy; [www.stonex.it](www.stonex.it)) to obtain a detailed topographic profile along the survey line.

The ERT dataset was collected with a Syscal Pro Switch 48 resistivity meter (IRIS Instruments, Orléans, France; www.iris-intruments.com). This was done with a dipole-dipole multi-skip acquisition scheme (Pavoni et al., 2023a), with reciprocal measurements and stacking ranging from 3 to 6 (Day-Lewis et al., 2008), for a total of 3542 measured data points. To ensure a good galvanic coupling, i.e., optimal contact resistances, and to obtain a high-quality dataset (Pavoni et al., 2022), conductive textile sachets, wet with salt water, were used as electrodes (Buckel et al., 2023; Bast et al., 2025).

The seismic data were collected with two Geode seismographs (Geometrics, San Jose, USA; http://www.geometrics.com), using vertical low-frequency geophones with a natural frequency of 4.5 Hz and a 20 kg sledgehammer as a seismic source. The source was moved from the first to the last geophone with a distance of six metres between each position, resulting in a total of 24 acquisition positions. At each position, the shot was repeated twice to stack the seismograms and enhance the signal-to-noise ratio.

## 3 MASW Method

Surface waves are seismic waves that travel along the Earth's surface, characterised primarily by dispersion, i.e., different frequencies propagate at different velocities (Everett, 2013). By analysing surface wave dispersion, it is possible to infer different mechanical properties of the medium through which the surface waves propagate (Socco et al., 2010). The depth of investigation of surface waves is associated with the seismic wavelength; a general rule of thumb is to consider one-third to one-half of the seismic wavelength of the lowest frequency component as the maximum penetration depth (Foti et al., 2015). Surface waves are also characterised by multi-modal propagation, meaning they can propagate in multiple modes simultaneously, including the fundamental mode and higher-order modes. The fundamental mode is the simplest form of wave propagation, with higher sensitivity near the surface, typically showing lower propagation velocities and higher amplitudes. Higher-order modes involve more complex sensitivity patterns with depth, can penetrate deeper layers, and usually exhibit higher velocities and lower amplitudes. However, the energy distribution of surface waves over different modes strongly depends on the subsurface conditions, and if higher modes with significant amplitude are present, special attention must be devoted to identifying the different modes (Boaga et al., 2013).

Surface wave analysis allows the retrieval of the dispersion relation (phase/group velocity versus frequency). In particular, the Multichannel Analysis of Surface Waves (MASW; Park et al., 1999) uses linear arrays to record the surface wave propagation from an active source in the time-space domain (seismogram). The acquisition setup is identical to SRT, but low-frequency geophones, having typically a natural frequency of 4.5 Hz, are essential for MASW surveys. The seismogram is converted into a frequency-wavenumber (f-k) or frequency-velocity (f-v) spectrum, where the energy maxima corresponding to the different modes are picked. Depth inversion is finally needed to retrieve a 1D Vs profile. Inversion is a non-linear ill-posed problem that can be solved deterministically using the linearized iterative least-squares approach (Herrmann, 1987), or with a stochastic search method, such as the neighbourhood algorithm (Sambridge, 1999). In both cases, some preliminary information is needed to define the starting model (deterministic approach) or the parameter space (stochastic approach).

The MASW method assumes homogeneous lateral conditions under the recording array. This condition is hardly met in nature, and when strong lateral heterogeneities are present, the complexity of the resulting spectra could challenge the picking process. For this reason, MASW is sometimes applied using moving windows. In this case, a quasi-2D Vs profile is retrieved, and smooth lateral velocity variations can be identified (Bohlen et al., 2004;

Boiero and Socco, 2011). The selection of the moving window length is crucial and requires preliminary testing:
a shorter window length causes an increase in lateral resolution but decreases the spectral resolution.

**4 ERT, SRT and MASW data processing, results and interpretation**


**4.1 ERT and SRT**

The ERT data processing was conducted using the open-source Python-based software *ResIPy*
(Blanchy et al., 2020), filtering the quadrupoles with reciprocal and stacking errors exceeding 5 %, which was
considered as the expected data error in the inversion modelling (Day-Lewis et al., 2008). This resulted in the
removal of 344 quadrupoles over 1324. The inverted resistivity model (Fig. 2a) was found in two iterations and
with a final RMS (Root-Mean Square) misfit of 1.17.
SRT data processing was performed with two open-source tools. Geogiga Front End Express v. 10.0, from
Geogiga Technology Corp. (https://geogiga.com/products/frontend/), was used for the picking of first arrivals,
while the C++/Python-based library pyGIMLi (Rücker et al., 2017) was used for data inversion. For each
seismogram, first arrivals were picked multiple times for the same shot and also considering reciprocal shots along
the array, in order to estimate the picking error (1 ms) to be used in the inversion process (Bauer et al., 2010). The
inverted P-wave velocity (Vp) model (Fig. 2b) was obtained after five iterations, with a final $\chi^2$ (chi-square) misfit
of 1.31.

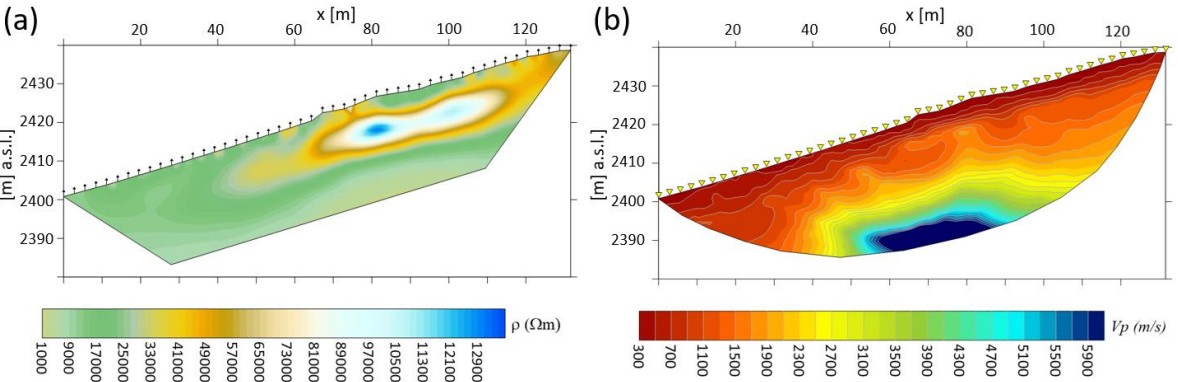

**Figure 2: (a) The inverted ERT section, after two iterations, has an expected data error of 5% and a final RMS misfit**
**of 1.17. The black markers along the surface indicate the positions of the electrodes. (b) The inverted P-wave velocity**
**model (SRT), after five iterations, has a picking error of 1 ms and a final $\chi^2$ misfit of 1.31. The yellow triangular**
**markers along the surface indicate the positions of the geophones.**
In the upper 4 - 5 m of the ground, electrical resistivity values are relatively high (~ 20 kΩm), and Vp values are
particularly low (< 600 m/s). This indicates a highly porous layer composed of blocks and debris with low fine
sediment content. Towards the front of the rock glacier (x < 40 m) and at greater depths, the electrical resistivity
decreases (< 10 kΩm), and the Vp values gradually increase, reaching 1200-1500 m/s at the bottom of the model.
Here, the substrate appears more heterogeneous, consisting of a mix of coarse debris and finer sediments. Towards
the upper section of the rock glacier lobe, at 4 - 5 m depth, resistivities increase (~ 40 kΩm) for 40 m < x < 60 m,
with an even sharper rise to values > 80 kΩm for x > 60 m. These values are typical for an ice-bearing frozen

layer (Hauck and Kneisel, 2008; de Pasquale et al., 2022). This layer extends to a depth of 10 - 12 m before resistivities clearly decrease to a few kΩm at the bottom of the model. In the Vp model, values increase at greater depths, with a steep gradient at ~ 20 m depth (50 < x < 80 m), where Vp Values reach 6000 m/s, indicating the bedrock. In the upper part of the section, between 4 - 5 m and 10 - 12 m depth, no typical Vp values of an ice-bearing frozen layer are reached (Vp > 2500 m/s, Hauck and Kneisel, 2008). Therefore, in this area, the ERT and SRT results are inconsistent: while the inverted resistivity model clearly indicates an ice-bearing layer, the Vp model shows a moderate increase, peaking at Vp values ~1500 m/s, likely corresponding to a liquid water-saturated layer.

## 4.2 MASW

The MASW analysis was performed using a moving window of 24 channels, striking a balance between spatial and spectral resolution. An offset-dependent mute was applied to those shot gathers that presented at least one source bounce as this could significantly impact the subsequent phase measurements. The time of occurrence of the source bounces was automatically identified through the auto-correlation of the near-offset traces. The mute was finally applied to the seismogram to mask the source bounce. Each shot gather was then Fourier transformed in both time and space to obtain the corresponding f-k spectrum, from which the fundamental mode was manually picked. The retrieved dispersion curves were depth inverted using *Dinver* (Wathelet, 2008), an open-source tool included in *Geopsy* (https://www.geopsy.org/; last access: 28 February 2025) that performs a stochastic inversion based on the neighbourhood algorithm (Sambridge, 1999). *Dinver* requires the definition of the model space with a fixed number of layers. We used a four-layer model and parameterized each layer with a wide range of seismic velocities and Poisson ratios, while keeping the density constant (Tab. 1). This choice was guided by the preliminary information we gained from ERT and SRT sections, that would indicate two to three layers, depending on the presence of permafrost, and a relatively shallow seismic bedrock. *Dinver* generates a multitude of random models within the model space and calculates for each of these models a dimensionless misfit between observed and modelled dispersion curves (Whatelet et al., 2004). The final model is characterised by the minimum misfit.

**Table 1: Parameter space used for the dispersion curve inversion with the open-source tool *Dinver* (Wathelet, 2008). Abbreviations: Vp: P-wave velocities, Vs: S-wave velocities, ρ: density.**

|         | Thickness [m] | Vp [m/s]    | Vs [m/s]   | Poisson ratio | ρ [kg/m3] |
|---------|---------------|-------------|------------|---------------|-----------|
| 1       | 2 - 12        | 400 - 1000  | 200- 500   | 0.2 - 0.45    | 1800      |
| 2       | 2 - 12        | 800 - 5000  | 500 - 2500 | 0.2 - 0.45    | 2000      |
| 3       | 2 - 12        | 800 - 5000  | 500 - 2500 | 0.2 - 0.45    | 2000      |
| Bedrock | Infinite      | 2400 - 6000 | 1200- 3000 | 0.2 - 0.45    | 2200      |

Figure 3 shows the results of the picking (Figs. 3c and 3d) and the Vs models (Figs. 3e and 3f) derived from the inversion of two dispersion curves. The first curve refers to a shot placed on the left side and the first 24 geophones (Fig. 3a), while the second curve relates to a shot on the right side and the last 24 geophones (Fig. 3b). Despite

the noisy character of the seismograms, where strong scattering is observed, the f-k spectra show coherent energy
and at least one mode of propagation is clearly recognisable, assumed to be the fundamental mode (Figs. 3c
and 3d). The two spectra show different frequency and wavenumber distributions, indicating different subsurface
conditions. The maximum penetration depth, which is approximately half of the wavelength, can be computed
from the minimum picked frequency, and it is about 15 m. The inversion results reveal a smooth increase of
velocity with depth in the left part of the section, i.e., towards the front of the rock glacier (Fig. 3e), while it clearly
highlights a shallow (5 m depth) high-velocity layer (2000 m/s) on the right side, i.e., the upper part of the rock
glacier (Fig. 3f). The high-velocity layer has a thickness of approximately 5 m. At a depth of 10 m, a clear and
sharp decrease in the velocity is observed. Good convergence is reached in the inversion down to the maximum
sensitivity of 15 m. Below this depth, results should be treated with caution. The lack of convergence manifests
as a wide velocity range with a similar misfit: most models in this depth range are equally plausible. Lower-
frequency data is needed to constrain the inversion at greater depths. It is important to note that the limited
frequency range characterising the picked dispersion curves is partly due to the loss of high frequency from
scattering and partly to the presence of a high-impedance boundary (the top of the bedrock in the left half of the
section and the top of the frozen layer on the right) that likely prevents most of the low-frequency energy from
penetrating below.

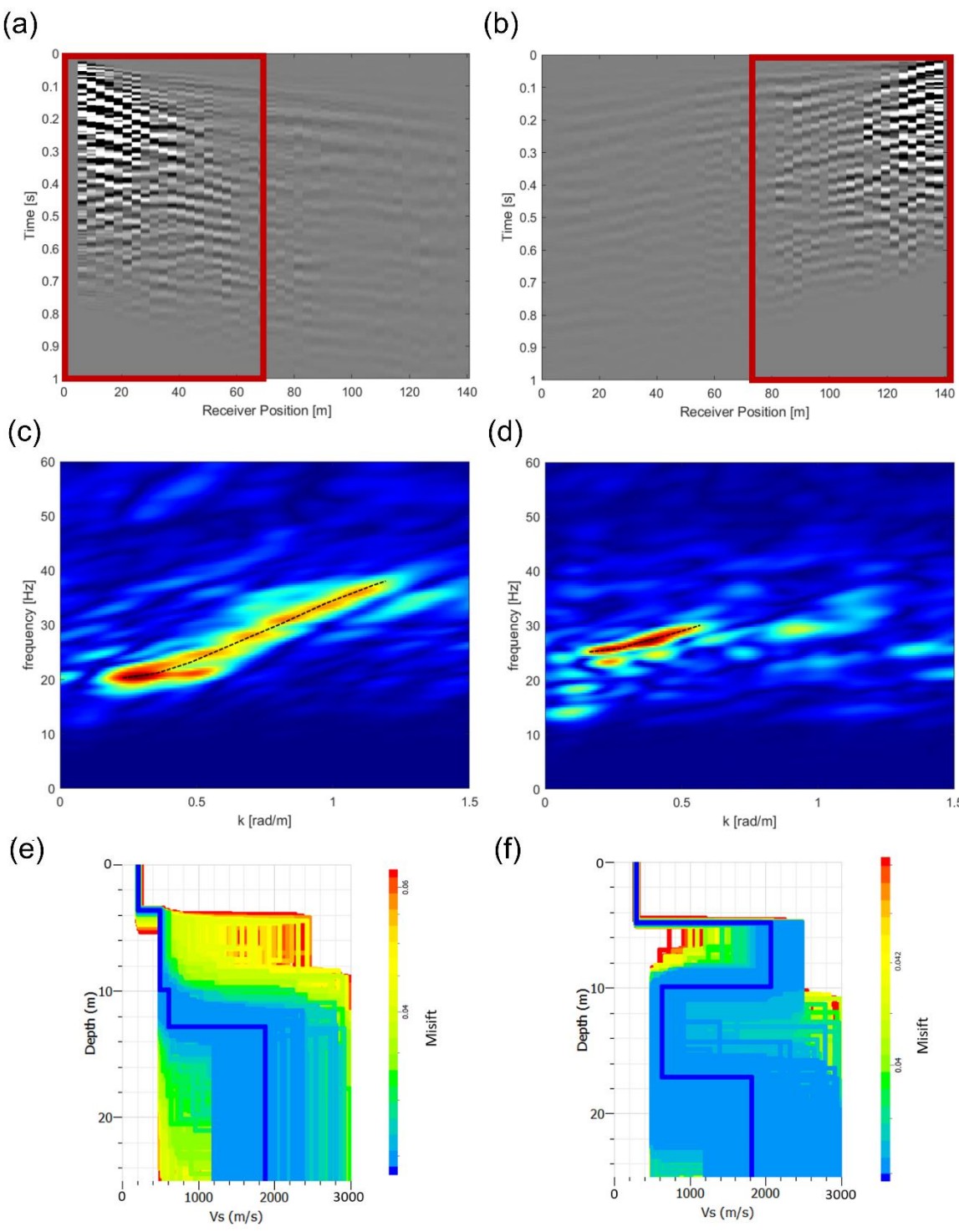

**Figure 3: (a)** Seismogram (grayscale) of the leftmost shot, where the red rectangle indicates the selected receivers for analysis. The offset-dependent mute effect is visible after 0.7 s, obscuring the source bounce. **(c)** Frequency-wavenumber (f-k) spectrum of the traces highlighted in (a), with the fundamental mode marked by a black dotted curve. The colours represent the seismic energy (low energy in cold colours / high energy in warm colours). **(e)** Depth inversion result of the picked dispersion curve, where colours represent different misfit values; the dark blue bold line signifies the final solution model, with a misfit of 0.02416. **(b), (d),** and **(f)** correspond to (a), (c), and (e), respectively, but for the rightmost shot. In this case, the misfit of the final solution model is 0.03797.

**4.3 Interpretation**

The obtained Vs models align well with the inverted resistivity section (Fig. 2a). The Vs values of the shallow (5 - 10 m depth) high-velocity (2000 m/s) layer observed in the right part of the section (Fig. 3f) are indeed consistent with the presence of an ice-bearing permafrost layer (Kuehn et al., 2024) that overlies a lower velocity layer of unfrozen sediments. Conversely, at depths of 5 m and below, the inverted SRT model indicates Vp values that are too low to support this conclusion, with a maximum value of 1500 m/s, which is characteristic of liquid water-saturated sediments. This suggests the presence of a supra-permafrost water layer, which can be commonly found in rock glacier environments where the frozen layer acts as an aquiclude (Krainer et al., 2007; Pavoni et al., 2023; Arenson et al., 2022, Jones et al., 2019). The ERT model does not resolve the presence of this (thin) water-saturated layer, likely considering the relatively large spacing of 3 m between the electrodes, nor does the MASW, which is sensitive to S-waves and thus insensitive to fluids. However, the ~ 1500 m/s P-wave velocities retrieved by the SRT method may indicate the presence of a (thin) water-saturated layer. In fact, such a layer may strongly attenuate body wave transmission and mask further impedance contrasts at depth (Pride et al., 2004). To assess the reliability of our subsurface hypothesis, we conducted both full-wave (FW) seismic modelling and synthetic ERT modelling (Chapter 5).

**5 Seismic and ERT Synthetic Modelling**

**5.1 Seismic synthetic modelling**

To verify the reliability of the obtained results, we generated synthetic seismograms based on a simplified subsurface model derived from the joint interpretation of ERT, SRT and MASW results. Synthetic shot gathers are compared to the real ones in terms of surface waves and first-arrival times.

Synthetic seismic data are generated using SW4 3.0 (Petersson and Sjögreen, 2023), which solves the seismic wave equations in Cartesian coordinates for 3D heterogeneous media (Sjögreen and Petersson, 2012; Zhang et al., 2021). The conceptual model for the simulation is shown in Fig. 4. The left part of the model is characterised by three main layers: (i) a 5 m-thick debris layer, (ii) a 12 m-thick layer of more compacted sediments and (iii) the bedrock. On the right side of the section, we included a 5 m-thick ice-bearing layer, and we hypothesised a 1 m-thick water-saturated layer above it. This model serves as a simplified representation of the assumed real subsurface, where clearly, the shape, thickness, and composition of the different layers are not regular and homogeneous. Moreover, it does not reproduce the small-scale heterogeneities in the model that are beyond the resolution of our field surveys. However, it represents the main structures highlighted by the MASW, ERT and SRT results, with the velocity and thickness of the different layers compatible with the results illustrated in Chapter 4.

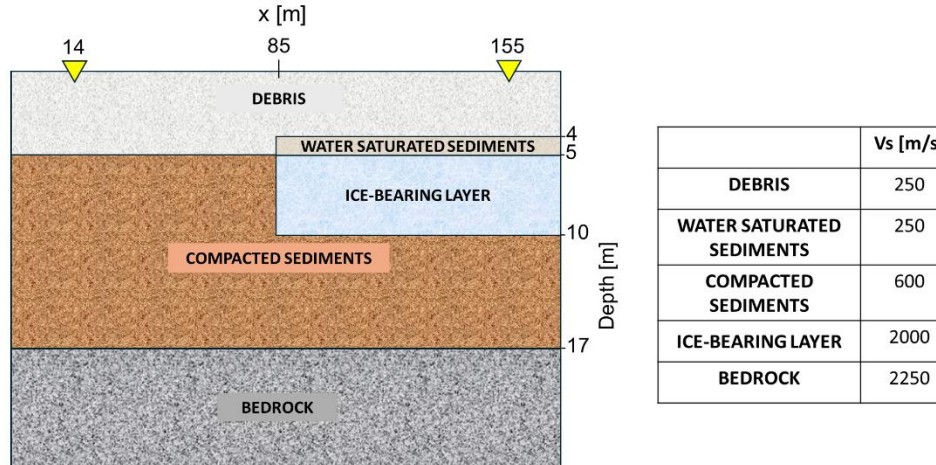

| | Vs [m/s] | Vp [m/s] | ρ [kg/m³] |
|---|---|---|---|
| **DEBRIS** | 250 | 625 | 1800 |
| **WATER SATURATED SEDIMENTS** | 250 | 1500 | 1800 |
| **COMPACTED SEDIMENTS** | 600 | 1500 | 2000 |
| **ICE-BEARING LAYER** | 2000 | 3500 | 2000 |
| **BEDROCK** | 2250 | 5000 | 2200 |


**Figure 4: Conceptual model used for the synthetic seismic modelling with the SW4 software (Petersson and Sjögreen,**
**2023). The two yellow triangles denote the first and the last geophones in the array. Abbreviations as in Tab. 1.**
The simulation domain is 170 x 30 x 30 m in the x, y and z directions. Absorbing boundaries were included in the
model to prevent the generation of reflections from the model edges, both at its bottom and laterally, while a free
surface condition was set at the top. The grid step used was 0.5 m, and the time step automatic setting was 0.87
ms, to comply with the stability criteria. The source was a vertical point load at the surface with central frequency
and maximum frequency of 15 Hz and 50 Hz, respectively. This choice was again imposed by the numerical
stability of the forward simulation. An array composed of 48 vertical receivers, with a spacing of 3 m, was placed
in the middle of the model (14 m ≤ x ≤ 155 m, y = 15 m) to reproduce the real case geometry. Two simulations
were run, corresponding to a shot on the left side of the array at the location of the first receiver and a shot on the
right side at the last receiver location.
Figures 5a and 5b show the synthetic shot gathers as grayscale plots. When compared to Figs. 3a and 3b, it is clear
how much noisier the field data are compared to the synthetic ones. This is the effect of scattering caused by the
boulders and coarse debris at the surface of the rock glacier. Consequently, the real f-k spectra (Figs. 3c and 3d)
are also noisier than the synthetic ones (Figs. 5c and 5d). However, the frequency and wavenumber distribution
of the fundamental mode in the modelled data is similar to the field observations. This is confirmed by comparing
the picking of modelled and real spectra (Figs. 5e and 5f). As highlighted in the scatterplots, the phase velocity
values obtained by sampling the fundamental mode in the synthetic spectra show a high correlation with the
corresponding values obtained from the field spectra ($R^2$ value ~ 0.99). Note that the comparison was made by
considering the phase velocity values obtained in the common frequency range in sampling the field spectrum
(Figs. 3c and 3d) and the synthetic spectrum (Figs. 5c and 5d), i.e., 20-35 Hz on the left side and 25-30 Hz on the
right side.

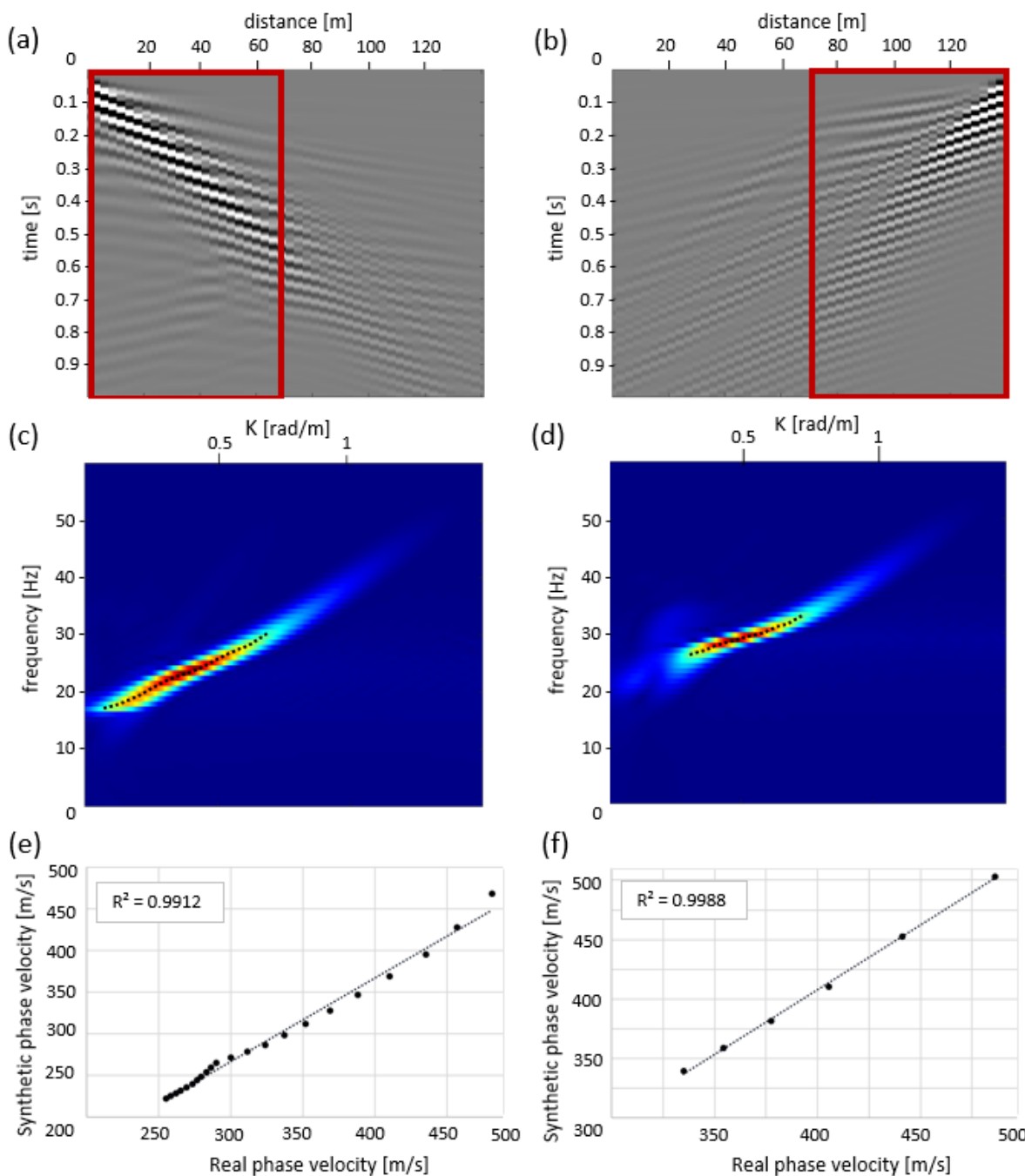

**Figure 5: (a) Synthetic seismogram (grayscale plot) of the left shot, where the red rectangle indicates the selected receivers for analysis. (c) Frequency-wavenumber (f-k) spectrum of the traces highlighted in (a), with the fundamental mode marked by a black dotted curve. (e) Scatterplot of the phase velocity picking obtained from the real (Fig. 3c) and synthetic spectrum (Fig. 5c). The black dotted lines show a simple linear regression line with corresponding $R^2$ values. (b), (d), and (f) correspond to (a), (c), and (e), respectively, but for the rightmost shot.**

First-arrival times picked on the modelled data are highly consistent with the field ones. Figure 6a shows the synthetic shot gathers (wiggle mode, normalized trace by trace) for sources on the left side of the geophone array, with the synthetic first arrivals (red lines) closely matching those in the field seismogram (Fig. 6c), as confirmed by a scatterplot and a high $R^2$ value (0.97; Fig. 6e). Similarly, for sources on the right side, synthetic (Fig. 6b) and field (Fig. 6d) shot gathers exhibit comparable first-arrival times (red lines), with a high $R^2$ value (0.95; Fig. 6f).

It is important to notice that synthetic first-arrival traveltimes were not merely modelled kinematically, but with a full-wave simulation which takes into account attenuation. In principle, kinematic modelling should generate traveltime curves whose slopes are compatible with the presence of the ice-bearing layer. Full-wave modelling is instead reproducing the attenuation effects of real data. Moreover, the low-frequency content of the source wavelet used for the simulation, imposed by the stability criteria, results in a rather low temporal resolution of first arrivals, which may generate uncertainties in picking comparable to the observed ones.

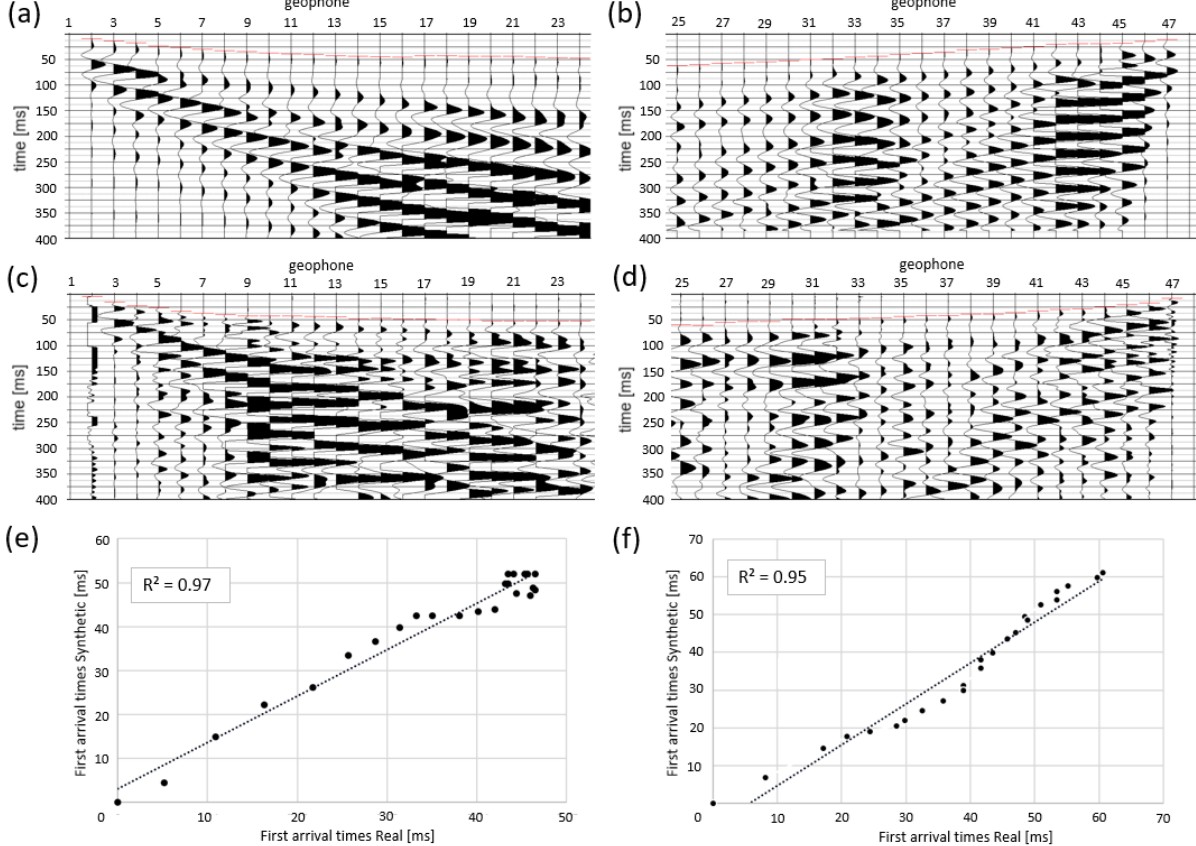

**Figure 6: Considering the conceptual model of Fig. 4a, (a) synthetic seismogram of the left shot plotted in wiggle mode (normalized trace by trace) and the picking (red lines in the traces) of first-arrival times for the first 24 traces; (c) field seismogram on the left shot plotted in wiggle mode and the picking (red lines in the traces) of first-arrival times for the first 24 traces; (e) scatterplot of synthetic first-arrival times and field ones for the left shot. The black dotted lines show a simple linear regression line with corresponding $R^2$ values. (b), (d), and (f) correspond to (a), (c), and (e), respectively, but for the rightmost shot.**

The good agreement between synthetic and field data regarding surface wave dispersion and first-arrival times, demonstrates the validity of the simple conceptual model presented in Fig. 4, which was used for the forward simulation. However, slight differences in the synthetic and field picking of the fundamental mode and first-arrival times may relate to the simplification of the synthetic model, which could not account for the highly complex topography and the heterogeneities of shape, thickness, and composition in the different layers.

**5.2 ERT synthetic modelling**

ERT synthetic modelling involves the numerical simulation of the electrical potential distribution in the subsurface based on a known resistivity model. This process requires solving Poisson's equation, which describes the behavior of the electric field generated by current injection through electrodes placed on the surface or in boreholes (Binley & Slater, 2020). In this study, the process was performed using the open-source software ResIPy (Blanchy et al., 2020) and the objective was to evaluate whether the electrode array and acquisition configuration used during the measurement campaign at the Flüela rock glacier provided sufficient resolution to detect a thin layer of water-saturated sediment overlying the permafrost. We hypothesize that this layer may have contributed to the attenuation of P-wave propagation at depth.

The synthetic modelling was based on the subsurface structure shown in Fig. 4, with electrical resistivity values assigned to each layer according to the inverted resistivity model derived from field data (Fig. 2a). Specifically, resistivities of 20 kΩ·m, 10 kΩ·m, 5 kΩ·m, and 100 kΩ·m were assigned to the surface debris layer, compact sediment, bedrock, and frozen layer, respectively (Fig. 7a). A representative value of 1 kΩ·m was assigned to the water-saturated sediment layer. In rock glacier environments, such layers can exhibit resistivities depending on factors such as material composition, water chemistry, and temperature. The assigned value is plausible particularly when the substrate consists of coarse, blocky debris with large pore spaces and low clay content, which tends to maintain relatively high resistivity even under saturated conditions (Hauck & Vonder Mühll, 2003; Hilbich et al., 2021). Additionally, if the pore water has low ionic content—as is typical of glacial meltwater— the resulting resistivity remains relatively high (Hauck, 2002).

The synthetic dataset was generated using a dipole–dipole multi-skip acquisition scheme identical to that employed in the field survey, with an array of 48 electrodes spaced 3 meters apart. A 5% noise level was added to the synthetic measurements, consistent with the estimated noise in the real dataset. The synthetic data were then inverted using the same parameters applied to the inversion of the real dataset, resulting in the resistivity model shown in Fig.7b. The result does not clearly reveal the presence of the thin water-saturated sediment layer overlying the frozen layer, confirming that the ERT survey conducted at the Flüela rock glacier site lacked the resolution and configuration necessary to resolve such a feature. This limitation is likely due to the relatively large electrode spacing.

Compared to the real electrical resistivity model (Fig. 2a), slight deviations can be observed, which can be attributed to the simplifications adopted in the conceptual model which does not account for the natural heterogeneity typically encountered in the field, including lateral and vertical variations in layer thickness, composition, and continuity. As in the seismic synthetic modelling, we assumed laterally homogeneous, planar layers and excluded surface topography, resulting in an idealized representation intended to enhance the theoretical detectability of the target layer.

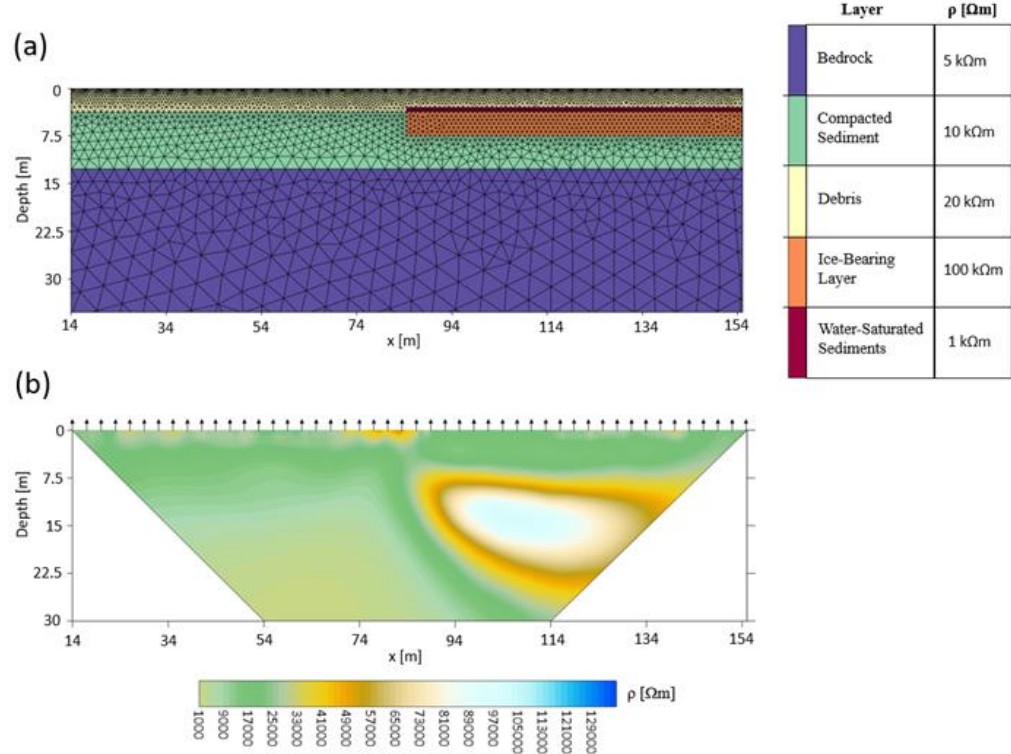

376

**Figure 7: (a) Conceptual model used for the synthetic ERT modelling; (b) Synthetic electrical resistivity model derived from forward modelling applied to the conceptual model presented in Fig. 7a.**

## 6 Discussion

### 6.1 Rock glacier subsurface model and rock glacier hydrology

Based on our presented ERT, SRT, MASW field data results, and ERT and FW seismic synthetic modelling, we constructed a subsurface model of the Flüela rock glacier (Fig. 8).

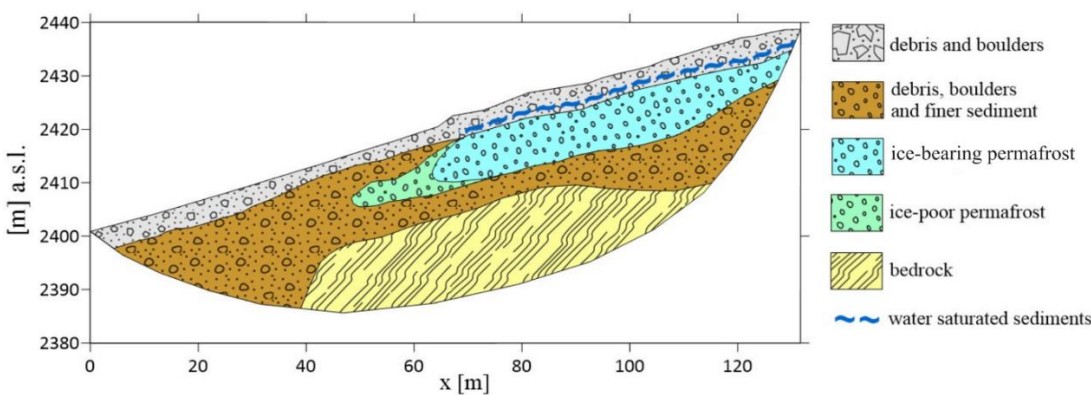

**Figure 8: Interpreted subsurface model of the Flüela rock glacier along the geophysical measuring line, derived from results of Electrical Resistivity Tomography (ERT), Seismic Refraction Tomography (SRT), Multichannel Analysis of Surface Waves (MASW), and Full-Wave (FW) seismic synthetic modelling.**

Four main units were identified. The uppermost layer, showing relatively high resistivity values (20 - 40 kΩm; Fig. 2a) and low seismic velocities ($V_p < 600$ m/s and $V_s \sim 250$ m/s; Fig. 2b and Figs. 3e-f), was interpreted as

mainly composed of debris and blocks, with high porosity (air) and poor regarding fine sediments. The deeper unfrozen sediment layer, with lower resistivities (< 10 kΩm) and relatively higher seismic velocities (Vp = 1200 - 1500 m/s and Vs ~ 500 m/s), was interpreted as a more heterogeneous compacted layer with both coarse and fine sediments. At the bottom of the model, the presence of bedrock was hypothesized, considering the sharp increase of Vp from 1200 - 1500 m/s to values > 3000 m/s (up to 6000 m/s), and of Vs values from ~ 500 m/s to ~ 2000 m/s at ~ 14 m depth for the left part of the section, and ~ 18 m depth for the right part. Considering that MASW applies a 1D approximation, the Vp model was mainly used to define the bedrock depth spatially. Finally, the high resistivity values (> 80 kΩm) identified in the right part of the resistivity model between ~ 5 m and ~ 10 m depth, corresponding to a sharp increase of Vs values (up to 2000 m/s), were interpreted as an ice-bearing permafrost layer. It should be noted that the high resistive layer also propagates beyond the middle of the array (50 < x < 65 m), but with lower values (~ 40 kΩm), probably linked to a decrease in the ice content or an increase in temperature.

Considering that an ice-rich layer typically acts as an aquiclude (Giardino et al., 1992; Krainer et al., 2007; Pavoni et al., 2023a; Arenson et al., 2022), we hypothesized the presence of a water-saturated layer above the permafrost. The ice-bearing layer is likely not detected in the Vp model because of (i) high picking uncertainties, due to the challenging environment (high level of scattering, wind noise) and (ii) the presence of a liquid-saturated layer that, in SRT studies, can obstruct energy transmission and mask additional impedance contrasts at depth (Carcione and Picotti, 2006; Picotti et al., 2007; Shi et al., 2024). This hypothesis can be further supported by the presence of a thin layer of fine-to-coarse sediments above a thicker, ice-bearing permafrost layer, as proposed by Boaga et al. (2024). These finer sediments are known to retain more water due to their smaller particle size, particularly if clay or silt is present (Hillel, 2003). However, without ground truthing, particularly drilling, we cannot obtain detailed subsurface information to confirm the exact structure and stratigraphy. Definitive statements regarding the ice and water content or the flow of water within the ice-bearing layer, such as intra-permafrost flow or the presence of taliks, cannot yet be made. Recent drilling in a rock glacier has revealed that areas identified as ice-rich using ERT and SRT methods can also contain significant amounts of liquid water and very fine sediments (personal observations by M. Phillips and A. Bast, SLF, 2024). Combining our methods with additional techniques could provide further insights into the hydrology of rock glaciers in the future. For example, Boaga et al. (2020) demonstrated that highly conductive anomalies in the subsurface, detected using Frequency Domain Electromagnetometry (FDEM) on a rock glacier, can indicate taliks or areas rich in liquid water.

**6.2 Advancements and challenges in using MASW for rock glacier characterisation**

Currently, the only two existing examples of MASW used for rock glacier characterisation are those by (i) Guillemot et al. (2021) at the Laurichard rock glacier, France, and (ii) Kuehn et al. (2024) at the Sourdough Rock Glacier, Alaska. However, in the first study, MASW was used in combination with other techniques to constrain a reference model of the unfrozen conditions for mechanical modelling of the rock glacier. In the second study, the aim of the study was to characterise the first few meters of a debris layer, achieved through a high-resolution seismic acquisition with sub-metre geophone spacing. Therefore, to our knowledge, the study presented here is the first successful application of MASW to derive structural information about rock glaciers, particularly

concerning the presence of the frozen layer. Indeed, surface wave analysis in periglacial environments is not straightforward. Surface wave penetration depends on the ability to generate low frequencies, which in turn requires heavy sources. The logistical constraints due to the high-mountain environments might hinder this method. Seismic datasets acquired in these contexts are also very noisy due to the scattering produced by debris and boulders and are highly attenuated, which reflects unclear modal distribution and narrow usable frequency bands. The mountainous environment may also affect data quality due to harsh weather conditions, particularly wind, and complex topography. Furthermore, rock glaciers are often highly heterogeneous media that vary significantly in both space and depth; complex 2D/3D structures could generate dispersion images that are difficult to interpret, which challenges data processing and interpretation. For this reason, the choice of the spatial window for the analysis should be made carefully to achieve the best lateral resolution without compromising spectral resolution. In the case of the Flüela rock glacier, the most natural choice was to perform MASW on the lower and upper halves of the line due to the relatively homogeneous conditions on each side. At locations with greater heterogeneity, selecting a suitable window length may be more difficult.

## 7 Conclusions

In this study, we highlighted the potential limitations of the SRT technique in accurately imaging ice-bearing layers in high-mountain rock glaciers, a limitation that may also apply to other permafrost environments. This limitation can be particularly relevant when a supra-permafrost water-saturated layer is present, acting as a preferential waveguide for seismic refractions and masking the underlying structures. Moreover, in these environments relatively high travel-time errors can further reduce the visibility of velocity contrasts. Another well-known limitation of the SRT method is its inability to image velocity inversions in the subsurface, such as an unfrozen sediment layer between the ice-bearing layer and the bottom bedrock.

As shown in our study, the surface wave analysis has the potential to overcome both of these limitations. Surface waves can be recorded simultaneously with the collection of seismic refraction data as long as low-frequency geophones are used for the acquisition. The analysis of surface wave dispersion in the frequency-wavenumber (f–k) spectrum, followed by the inversion of dispersion curves, enables the retrieval of Vs profiles, which are insensitive to the liquid phase (i.e., they are not affected by the presence of a supra-permafrost water-saturated layer). Moreover, surface wave dispersion analysis can retrieve velocity inversions with depth and resolve the presence of a low-velocity layer between high-velocity layers. This method is also less sensitive to random seismic noise due to scattering and external noise sources, which can interfere with the accurate picking of first-time arrivals in SRT analysis, providing an alternative and reliable solution for the analysis of seismic datasets affected by relatively high noise levels. At the Flüela rock glacier, the dispersion images of the left and right sides of the seismic section look different in terms of frequency content and velocity distribution. The Vs profile produced by the inversion of the right-side dispersion curve clearly shows an increase of velocities at 5 m depth, attributed to the ice-bearing layer, and a decrease at about 10 m, compatible with the presence of unfrozen sediments. This demonstrates the effectiveness of MASW for imaging the ice-bearing layer and the underlying unfrozen sediments, even in the presence of a supra-permafrost water layer and with a relatively noisy dataset, as well as the potential to retrieve the thickness of the ice-bearing layer to support the ERT findings.

In the future, we plan to implement the MASW technique across various locations, particularly where we have
borehole information on the subsurface stratigraphy, to further validate our findings. Additionally, we aim to
enhance the surface wave analysis by incorporating passive seismic data, such as ambient seismic noise captured
by seismic nodes, to extend our depth of penetration beneath the ice-rich layer to the seismic bedrock.
We recommend using low-frequency geophones and appropriate heavy sources whenever possible when
collecting SRT profiles. This approach will enable complementary MASW analysis and provide valuable
experience, which will undoubtedly benefit mountain permafrost research and enhance our understanding of ice
and water content in mountain permafrost, i.e., mountain permafrost hydrology.
*Data Availability Statement*: The datasets used to obtain the results presented in this work are available at the
open-source repository https://researchdata.cab.unipd.it/id/eprint/1748. Furthermore, the ERT datasets will also
be included in the International Database of Geoelectrical Surveys on Permafrost (IDGSP).
*Author contribution:* IB and MP developed the concept of the study; MP, JB, and AB collected the data; MP
performed the data processing of ERT and SRT data; IB and SJGT performed the MASW analysis; all authors
contributed to the interpretation of the results, writing and editing of the manuscript.
*Competing interests:* The contact author has declared that none of the authors has any competing interests.
*Acknowledgements:* The authors want to thank Dr Marcia Phillips for her valuable help and for pre-reviewing the
manuscript.
*Financial support:* The project "Cold spot" is part of the excellence program: "The Geosciences for Sustainable
Development" project (Budget Ministero dell'Università e della Ricerca - Dipartimenti di Eccellenza 2023–2027
C93C23002690001).

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
