# Peer review of "Multichannel Analysis of Surface Waves (MASW) for the"

_EGUsphere, 2025_

## Referee Comment (RC2)

[referee-annotated manuscript omitted]

---

## Author Comment (AC1)

**RC1**: ['Comment on egusphere-2025-962'](), Wojciech Dobiński, 19 May 2025
The work submitted for evaluation presents interesting and original research material obtained from field studies conducted using complementary ERT and seismic methods used for decades in studies on the occurrence of permafrost.

The area selected for the study is already very well-known from similar permafrost studies, of which a great many have been conducted in this area since the 1970s. Neither the choice of methods nor the choice of the area is therefore particularly original and rather fits into the trend of research conducted for many years.

ERT and seismic refraction are certainly the most traditionally used geophysical methods for characterizing rock glaciers (Hauck and Kneisel, 2008). However, the potential of the Multichannel Analysis of Surface Waves (MASW) has never been fully explored to characterize high mountain permafrost environments like those in the European Alps. To our knowledge, the only study on this subject published in the scientific literature is Kuehn et al. (2024), where, the MASW method only leads to the characterization of the rock glaciers' active layer. Additionally, Guillemot et al. (2021) used MASW in combination with other techniques to constrain a reference S-wave velocity distribution for the unfrozen conditions at the Laurichard rock glacier, with the real purpose of tackling seasonal variability using ambient seismic noise.

In our work, the MASW method plays a central role in fully reconstructing the structure of the Flüela rock glacier (active layer, frozen layer, unfrozen basal till, and bedrock). MASW overcomes the limitations of SRT, which does not reveal velocity inversions (thus, it does not allow to define the thickness of the frozen layer) and may not reveal the characteristic velocities of the frozen layer when, according to our interpretation, a thin saturated supra-permafrost layer exists. Consequently, relying solely on ERT and SRT models could challenge the interpretation and understanding of the rock glacier structure. On the other hand, MASW provides more reliable results, leading to a more accurate model of the subsurface.

Our novel and original findings, open new perspectives on the possible use of MASW for the permafrost characterization in rock glaciers. Through this work, we aim to encourage the mountain permafrost research community to collect active seismic data using low-frequency geophones (4.5 Hz) whenever possible. Although this data is typically used for refraction analysis, recording it with low-frequency sensors also enables the application of the MASW method. This dual use can help minimize interpretative ambiguities or even improve the overall interpretation, as we successfully demonstrated in our case study.

The article can be divided quite clearly into a part concerning permafrost and a part concerning methods. The authors focus strongly on the latter, because its specific application brings the most interesting scientific result. However, I will start with the issue related to permafrost.

Here, a very sensible approach to permafrost in general is worth noting, in which the authors avoid terms such as 'permafrost creep', 'ice-rich' or 'ice-poor permafrost', 'permafrost hydrology', etc. This is a big advantage for the work, because these very simplified and in fact incorrect terms are still and quite often used in permafrost research. It should be emphasized here that for many years there has been a general agreement regarding the definition of permafrost, which describes it as a state of the ground. Therefore, since permafrost is a thermal state, it is impossible to assign a material expression to it. The authors seem to understand this well by avoiding incorrect terms, but they do not do it consistently and unfortunately use some incorrect terms interchangeably. I have noted some cases of such use in the reviewed work, which I am sending as an attachment and which is part of the review.

Indeed, using the correct terminology is essential in science and, therefore, in permafrost and geophysical research. We will ensure consistency with the terminology and modify any incorrectly used terms according to the glossary of permafrost and related ground-ice terms published by the International Permafrost Association (IPA).

The introductory part also lacks at least a short critical review of geophysical studies of permafrost in the studied region and a short review of the application of the MASW method in the study of permafrost in mountain and Arctic environments. Such a text would allow for better highlighting the achievement that the authors describe in the work. See for example:

Kula D, Olszewska D, Dobiński W, Glazer M, 2018. Horizontal-to-vertical spectral ratio variability in the presence of permafrost. Geophysical Journal International 214, 1, 219-231

We agree on the need to extend the literature review to other methods using ambient vibrations and surface waves, and to permafrost studies in general. However, site conditions in the Arctic generally do not typically present the challenges of rock glaciers in terms of rough topography, presence of very large boulders, lateral heterogeneity etc. that represent the main obstacles for the MASW application. In fact, several studies already exist in the literature where MASW is used, often in combination with ERT, to characterize permafrost in Arctic regions (Glazer et al. 2020, Liu et al. 2022, Tourei et al. 2024), and we will reference to them in the new manuscript. If we strictly refer to Alpine rock glaciers, to our knowledge, only two MASW case studies exist in literature (Kuehn et al., 2024; Guillemot et al. 2021), that we already cited in the original manuscript and that we already commented on above.

The proposal to determine the presence of permafrost based on the results of original studies is very interesting, because the lack of agreement between ERT and seismic results is very well filled by MASW and this is an original and important result of these studies, most worthy of publication and testing also in other conditions and by other researchers.

However, I have the impression that the article focuses too much on methodological issues, which makes the article more engineering than scientific in nature. While characterizing the methodology and the results of empirical research well, it leaves the proposed models of permafrost occurrence without further discussion. As I noted at the beginning, we know that many similar research works have been carried out in this area since the 1970s. Therefore, in my opinion, it is also important to compare the obtained results with those that are already in scientific circulation. Against this background, the empirical model of permafrost occurrence constructed by the authors will be more credible, more universal and ready to be applied also in other permafrost occurrence environments. This may cause the work to become more universal and more widely cited in the scientific community.

We submitted to the special issue *"Emerging geophysical methods for permafrost investigations: recent advances in permafrost detecting, characterizing, and monitoring"*. For this reason, we believe that all technical details provided in the manuscript regarding the geophysical methods are needed to address both the geophysics community and the permafrost community.

As for the previous research, the preliminary investigations that we describe in section 2 primarily relate to a different area of the Flüelapass rock glacier, where the conditions are likely very different. The only geophysical studies in the lower tongue were conducted by Haeberli (1975), Boaga et al. (2024) and Bast et al. (2025). We refer to all three publications in the manuscript. The work of Haeberli (1975), conducted 50 years ago, involves refraction seismic data only, which—as demonstrated in our study—can lead to ambiguities in result interpretation, especially when not combined with the ERT method. Furthermore, we do not know the exact location of the geophysical investigations. The publications of Boaga et al. (2024) and Bast et al. (2025) refer to the same profile as we used for our presented work. The authors focus primarily on methodological developments rather than on a comprehensive characterization of the rock glacier itself. There are no boreholes on the rock glacier, meaning there is no direct information (e.g., on temperature or the stratigraphy) about its internal structure. In conclusion, unfortunately no previous models or borehole data exist for the lower tongue of the Flüelapass rock glacier, which could prove or disprove our findings. That considered, our work has significantly improved both the understanding and the reliability of the structural model of the Flüelapass rock glacier.

**References**

Bast, A., Pavoni, M., Lichtenegger, M., Buckel, J., and Boaga, J.: The use of textile electrodes for electrical resistivity tomography in periglacial, coarse blocky terrain: a comparison with conventional steel electrodes, Permafrost and Periglacial Processes, 36(1), 110-122, https://doi.org/10.1002/ppp.2257, 2025.

Boaga, J., Pavoni, M., Bast, A., and Weber, S.: Brief communication: On the potential of seismic polarity reversal to identify a thin low-velocity layer above a high-velocity layer in ice-rich rock glaciers, The Cryosphere, 18, 3231–3236, https://doi.org/10.5194/tc-18-3231-2024, 2024.

Glazer, M., Dobiński, W., Marciniak, A., Majdański, M., and Błaszczyk, M.: Spatial distribution and controls of permafrost development in non-glacial Arctic catchment over the Holocene, Fuglebekken, SW Spitsbergen, Geomorphology, 358, 107128, https://doi.org/10.1016/j.geomorph.2020, 2020.

Guillemot, A., Baillet, L., Garambois, S., Bodin, X., Helmstetter, A., Mayoraz, R., and Larose, E.: Modal sensitivity of rock glaciers to elastic changes from spectral seismic noise monitoring and modelling, The Cryosphere, 15, 501–529, https://doi.org/10.5194/tc-15-501-2021, 2021.

Haeberli, W.: Untersuchungen zur Verbreitung von Permafrost zwischen Flüelapass und Piz Grialetsch (Graubunden), Mitteilung der Versuchsanstalt für Wasserbau, Hydrologie und Glaziologie, ETH Zurich, Zurich, 1975.

Hauck, C., and Kneisel, C.: Applied Geophysics in Periglacial Environments, Cambridge University Press., 2008.

Kuehn, T., Holt, J. W., Johnson, R., and Meng, T.: Active seismic refraction, reflection, and surface wave surveys in thick debris-covered glacial environments, Journal of Geophysical Research: Earth Surface, 129, e2023JF007304, https://doi.org/10.1029/2023JF007304, 2024.

Liu, H., Maghoul, P., and Shalaby, A.: Seismic physics-based characterization of permafrost sites using surface waves, The Cryosphere, 16, 1157–1180, https://doi.org/10.5194/tc-16-1157-2022, 2022.

Tourei, A., Ji, X., Rocha dos Santos, G., Czarny, R., Rybakov, S., Wang, Z., Hallissey, M., Martin, E. R., Xiao M., Zhu, T., Nicolsky, D., and Jensen, A.: Mapping Permafrost Variability and Degradation Using Seismic Surface Waves, Electrical Resistivity, and Temperature Sensing: A Case Study in Arctic Alaska, Journal of Geophysical Research: Earth Surface, 129, e2023JF007352, https://doi.org/10.1029/2023JF007352, 2024.

---

## Author Comment (AC2)

**RC2**: 'Comment on egusphere-2025-962', Anonymous Referee #2, 12 Jun 2025

This paper presents a multi-method geophysical campaign utilizing Electrical Resistivity Tomography (ERT), Seismic Refraction Tomography (SRT), and Multichannel Analysis of Surface Waves (MASW) to characterize the subsurface of the Flüela rock glacier. The study effectively demonstrates the complementary strengths of MASW in permafrost environments, particularly in overcoming some limitations of conventional SRT, such as issues with velocity inversions. The comparison of synthetic seismic models with field data is a valuable aspect of the work, corroborating the authors' interpretations.

The manuscript is generally well-structured and clear. The methodology is adequately described, and the figures are informative. The application of MASW in this challenging high-mountain environment is a significant contribution to permafrost research.

**Main Comment:**

My primary concern revolves around the interpretation of the ERT results concerning the hypothesized thin, water-saturated layer. While it is acknowledged that ERT sensitivity decreases with depth, it is generally expected that a **1-meter thick layer near the surface** should be resolvable with a **3-meter electrode spacing**. Furthermore, since you are considering a **more conductive layer** (interpreted as water-saturated sediment), the ERT method should exhibit **heightened sensitivity** to its presence.

It would be highly valuable to include **synthetic ERT modeling** to illustrate the expected response to such a thin, low-resistivity layer at the proposed depth (around 4m, based on Figure 4). A synthetic model would help clarify whether the observed field ERT data aligns with the theoretical detectability of such a feature, given the acquisition parameters and the assumed resistivity contrast. This would strengthen the argument for why the ERT model does not clearly resolve this layer despite its potential conductivity.

We thank Reviewer2 for his/her comment, that gave us the opportunity to further investigate the sensitivity of our system (electrode array and acquisition configuration) to the hypothesized structures, in particular to the thin supra-permafrost water-saturated sediment layer. To do so, we performed a forward modelling of ERT using the open-source software ResIPy (Blanchy et al., 2020). The forward model was based on the subsurface structure shown in Figure 4 of the manuscript, with electrical resistivity values assigned to each layer according to the inverted resistivity model derived from field data (Figure 2a of the manuscript). Specifically, resistivities of 20 kΩ·m, 10 kΩ·m, 5 kΩ·m, and 100 kΩ·m were assigned to the surface debris layer, compact sediment layer, bedrock, and frozen layer, respectively (Figure S1 of the Supplementary Material). A representative value of 1 kΩ·m was assigned to the water-saturated sediment layer; as usual in rock glacier environments for such layer (depending on factors such as material composition, water chemistry, and temperature). This value is plausible particularly when the substrate consists of coarse, blocky debris with large pore spaces and low clay content, which reduces electrical conductivity even under saturated conditions (see Hauck & Vonder Mühll, 2003; Hilbich et

al., 2021). Additionally, if the pore water has low ionic content—as is typical of meltwater—the resulting electrical conductivity remains low, yielding higher resistivity values (Hauck, 2002). Cold yet unfrozen conditions, or partially saturated porous media, may also lead to resistivities within this range.

The synthetic dataset was generated using a dipole–dipole multi-skip acquisition scheme identical to that employed in the field survey, with an array of 48 electrodes spaced 3 meters apart. A 5% noise level was added to the synthetic measurements, consistent with the estimated noise in the real dataset. The synthetic data were then inverted using the same parameters applied to the inversion of the real dataset, resulting in the resistivity model shown in Figure S2 (Supplementary Material).

The result does not clearly reveal the presence of the thin water-saturated sediment layer overlying the frozen layer, confirming that the ERT survey conducted at the Flüelapass rock glacier lacked the resolution and configuration necessary to resolve such a feature. This limitation is likely due to the relatively large electrode spacing.

Compared to the real electrical resistivity model (Figure 2a of the manuscript), slight deviations can be observed, which can be attributed to the simplifications adopted in the conceptual model (Figure 4 of the manuscript and Figure S1 of the Supplementary Material). The conceptual model used for the synthetic simulation does not account for the natural heterogeneity typically encountered in the field, including lateral and vertical variations in layer thickness, composition, and continuity. As in the seismic forward modeling, we assumed laterally homogeneous, planar layers and excluded surface topography, resulting in an idealized representation intended to enhance the theoretical detectability of the target layer.

Overall, this is a well-executed study that provides important insights into the internal characteristics of rock glaciers. Addressing the main comment with additional synthetic modeling would significantly enhance the clarity and robustness of the ERT interpretation.

We thank Reviewer 2 for his/her clear and insightful review. We hope that our additional analyses with synthetic data were thorough and addressed all concerns regarding our interpretation.

**References**

Blanchy, G., Loke, M. H., Ogilvy, R., & Meldrum, P. (2020). ResIPy: A Python-based GUI for 2D/3D resistivity modeling and inversion. Journal of Open Source Software, 5(54), 2432. https://doi.org/10.21105/joss.02432.

Hauck, C. (2002). Frozen ground monitoring using DC resistivity tomography. Geophysical Research Letters, 29(21), 2016. https://doi.org/10.1029/2002GL014995.

Hauck, C., & Vonder Mühll, D. (2003). Detecting seasonal changes in permafrost using geophysical methods. Permafrost and Periglacial Processes, 14(3), 213–222. https://doi.org/10.1002/ppp.451.

Hilbich, C., Fuss, C., Mollaret, C., Hauck, C., & Hoelzle, M. (2021). Multi-decadal geophysical monitoring of permafrost evolution in mountain terrain – The PACE legacy. The Cryosphere, 15(11), 5121–5145. https://doi.org/10.5194/tc-15-5121-2021.

**Supplementary Material**

**ERT Synthetic (Forward) Modeling**

Forward modeling in Electrical Resistivity Tomography (ERT) involves the numerical simulation of the electrical potential distribution in the subsurface based on a known resistivity model. This process requires solving Poisson's equation, which describes the behavior of the electric field generated by current injection through electrodes placed on the surface or in boreholes (Binley & Slater, 2020). Forward modeling is a crucial step in the ERT workflow, as it allows for the prediction of the theoretical response of the subsurface for a given resistivity distribution and electrode configuration. It is commonly used to test the effectiveness of specific electrode arrays, assess the system's sensitivity to subsurface resistivity variations, and validate the quality of inversion results (Loke et al., 2003; Binley & Kemna, 2005).

In this study, forward modeling of ERT was performed using the open-source software ResIPy (Blanchy et al., 2020). The objective was to evaluate whether the electrode array and acquisition configuration used during the measurement campaign at the Flüelapass rock glacier provided sufficient resolution to detect a thin layer of water-saturated sediment overlying the permafrost. We hypothesize that this layer may have contributed to the attenuation of P-wave propagation at depth.

The forward model was based on the subsurface structure shown in Figure 4 of the manuscript, with electrical resistivity values assigned to each layer according to the inverted resistivity model derived from field data (Figure 2a of the manuscript). Specifically, resistivities of 20 kΩ·m, 10 kΩ·m, 5 kΩ·m, and 100 kΩ·m were assigned to the surface debris layer, compact sediment layer, bedrock, and frozen layer, respectively (Figure S1). A representative value of 1 kΩ·m was assigned to the water-saturated sediment layer. In rock glacier environments, such layers can exhibit resistivities on the order of 1000 Ω·m, depending on factors such as material composition, water chemistry, and temperature. This value is plausible particularly when the substrate consists of coarse, blocky debris with large pore spaces and low clay content, which reduces electrical conductivity even under saturated conditions (Hauck & Vonder Mühll, 2003; Hilbich et al., 2021). Additionally, if the pore water has low ionic content—as is typical of glacial meltwater—the resulting electrical conductivity remains low, yielding higher resistivity values (Hauck, 2002). Cold yet unfrozen conditions, or partially saturated porous media, may also lead to resistivities within this range.

[Figure]

*Figure S1: Conceptual model used for the synthetic ERT modelling.*

The synthetic dataset was generated using a dipole–dipole multi-skip acquisition scheme identical to that employed in the field survey, with an array of 48 electrodes spaced 3 meters apart. A 5% noise level was added to the synthetic measurements, consistent with the estimated noise in the real dataset. The synthetic data were then inverted using the same parameters applied to the inversion of the real dataset, resulting in the resistivity model shown in Figure S2. The color scale used corresponds to that of the electrical resistivity model obtained from the real data, presented in Figure 2a of the manuscript.

[Figure]

*Figure S2: Synthetic electrical resistivity model derived from forward modeling applied to the conceptual model presented in Figure S1.*

As shown in Figure S2), the result does not clearly reveal the presence of the thin water-saturated sediment layer overlying the frozen layer, confirming that the ERT survey conducted at the Flüelapass rock glacier site lacked the resolution and configuration necessary to resolve such a feature. This limitation is likely due to the relatively large electrode spacing.

Compared to the real electrical resistivity model (Figure 2a of the manuscript), slight deviations can be observed, which can be attributed to the simplifications adopted in the conceptual model (Figure 4 of the manuscript and Figure S1). The conceptual model used for the synthetic simulation does not account for the natural heterogeneity typically encountered in the field, including lateral and vertical variations in layer thickness, composition, and continuity. As in the seismic forward modeling, we assumed laterally homogeneous, planar layers and excluded surface topography, resulting in an idealized representation intended to enhance the theoretical detectability of the target layer.

**Ray density in Seismic Refraction Tomography (SRT)**

Figure S3 shows the Vp model obtained through Seismic Refraction Tomography (SRT) together with the computed ray paths. In the first half of the model domain (0 < x < 60 m, where we assume the absence of the thin water-saturated layer), ray coverage is well distributed both at the shallow and intermediate depths. Conversely, in the second half of the section (x > 60 m, where we hypothesize the presence of the saturated layer above the permafrost), the majority of rays are concentrated within the near-surface portion of the model, with only a limited number of rays penetrating to deeper levels.

[Figure]

*Figure S3: Vp model obtained through seismic refraction tomography (SRT), together with the computed ray paths.*

**References**

Blanchy, G., Loke, M. H., Ogilvy, R., & Meldrum, P. (2020). ResIPy: A Python-based GUI for 2D/3D resistivity modeling and inversion. Journal of Open Source Software, 5(54), 2432. https://doi.org/10.21105/joss.02432.

Binley, A., & Kemna, A. (2005). DC resistivity and induced polarization methods. In Rubin, Y. & Hubbard, S.S. (Eds.), Hydrogeophysics (pp. 129–156). Springer. DOI: 10.1007/1-4020-3102-5.

Binley, A., & Slater, L. (2020). Resistivity and Induced Polarization: Theory and Practice. Cambridge University Press. DOI: 10.1017/9781108685955.

Hauck, C. (2002). Frozen ground monitoring using DC resistivity tomography. Geophysical Research Letters, 29(21), 2016. https://doi.org/10.1029/2002GL014995.

Hauck, C., & Vonder Mühll, D. (2003). Detecting seasonal changes in permafrost using geophysical methods. Permafrost and Periglacial Processes, 14(3), 213–222. https://doi.org/10.1002/ppp.451.

Hilbich, C., Fuss, C., Mollaret, C., Hauck, C., & Hoelzle, M. (2021). Multi-decadal geophysical monitoring of permafrost evolution in mountain terrain – The PACE legacy. The Cryosphere, 15(11), 5121–5145. https://doi.org/10.5194/tc-15-5121-2021.

Loke, M. H., Acworth, I., & Dahlin, T. (2003). A comparison of smooth and blocky inversion methods in 2D electrical imaging surveys. Exploration Geophysics, 34(3), 182–187. DOI: 10.1071/EG03182.

---

## Author Comment (AC3)

**RC3**: 'Comment on egusphere-2025-962', Anonymous Referee #3, 07 Jul 2025

The submitted work proposes the use of Multichannel Analysis of Surface Waves (MASW) in conjunction with Seismic Refraction Tomography (SRT) and Electrical Resistivity Tomography (ERT) for permafrost characterization within a rock glacier. It effectively demonstrates the benefits of designing seismic surveys that can be processed not only for SRT but also for MASW. The complementary information provided by MASW is particularly valuable in the context of mountain permafrost studies, where it helps address the limitations of SRT and thus resolves structural discrepancies with ERT, ultimately leading to a more accurate understanding of subsurface composition.

The paper is generally well-structured and presents the methodology and materials in a clear and concise manner. I have just a couple of minor comments that I would like to raise.

A somewhat misleading statement in the manuscript is that the open-source library pyGIMLi was used for the processing of the SRT data. As pyGIMLi does not currently support first-arrival picking, it can be assumed that a different software package or custom code was used for this step. It would be helpful if the authors clarified which tool was employed for picking the travel times.

Indeed, the first break picking has been performed with the free software Geogiga Front End Express v. 10.0, from Geogiga Technology Corp. (https://geogiga.com/products/frontend/). We will add this information to the updated manuscript.

Furthermore, the use of 0 m/s as the lower bound in the colorbar of the SRT results shown in Figure 2(b) is questionable, as no real material exhibits a P-wave velocity of zero. It would be more appropriate to set a lower limit that reflects the minimum physically meaningful or measured velocity in the dataset.

The colorscale has been readapted with a more realistic lower bound (300 m/s; see figure below). However, the appearance of the section does not change dramatically, and the same structures are visible.

[Figure]

Regarding the SRT imaging result, both the synthetic and field data experiments suggest the presence of a thin, water-bearing layer above the ice lens, with no critical P-wave refractions observed beneath it. In this context, it may be more appropriate to display the actual ray coverage instead of what appears to be a convex hull surrounding the resolved model domain. This would provide a clearer indication of which regions are sensitive to the data. In particular, I would assume that the area at and below the interpreted ice body has poor/no coverage and therefore, variations in the model in these regions likely only show the starting model. These areas should perhaps not be emphasized in the comparison and interpretation of the results.

We will include in the supplementary material the Vp model together with the computed ray paths (Figure S3 of the supplementary material, shown below). As the figure illustrates, in the first half of the model domain (0 < x < 60 m, where we assume the absence of the thin water-saturated layer), ray coverage is well distributed both at the shallow and intermediate depths. Conversely, in the second half of the section (x > 60 m, where we hypothesize the presence of the saturated layer above the permafrost), the majority of rays are concentrated within the near-surface portion of the model, with only a limited number of rays penetrating to deeper levels. Considering that Vp values have been calculated across all model elements—even in zones exhibiting sparse ray coverage—we prefer to maintain Figure 2 as presented in the manuscript, with the concave mask delineated by the propagation limits of the outermost and deepest rays.

[Figure]

Another comment relates to the reliability of the MASW results, particularly given the narrow frequency range in the presumed ice-rich area, the low velocities observed in the extracted dispersion curves, and the apparent variability —and thus uncertainty — of the S-wave velocity models below 10-15 meters. As the authors note themselves, model variations beneath this depth should be treated with caution as they are likely not constrained by the data anymore. Hence, I wonder why the fourth layer was considered for the inversion at all. I think it would also be helpful to add further information on the initial model parameter space definition, i.e., what lead to the choice of number of layers, thickness distributions (e.g., was this guided by site-specific information based on the

other geophysical methods or prior investigations, etc.) and whether different set-ups were tested that lead to similar results.

Dinver uses a stochastic approach for inverting the dispersion curves. This means there is no initial model that could bias the final result. The only user-dependent parameters are the number of layers and the ranges of physical parameters in which each layer may move during the inversion process. We intentionally used very wide ranges for both thicknesses and velocities, so that the algorithm could freely explore a wide range of subsurface models, also allowing velocity inversions with depth. The choice of the number of layers was dictated by the preliminary information we had from ERT and SRT sections, that would indicate two to three layers, depending on the presence of permafrost, and a relatively shallow seismic bedrock (this information will be added to the new manuscript). It is true that the low velocities shown by the spectra already reveal the lack of sensitivity to the bedrock, but we preferred to use a realistic model parametrization based on a priori geological/geophysical information and then make an a-posteriori evaluation of the sensitivity based on the inversion results.

We think that the seismograms and their relative spectra further corroborate the results of the inversion, since not only the dispersion characteristics but also the energy distribution over the different frequencies well reproduce the experimental spectra.

Additionally, the final misfit values and error bounds used for the MASW inversion should be added in the text. For matters of consistency, the authors could also consider adding the staring conditions for the deterministic SRT and ERT inversions.

The misfit computed by Dinver depends on the number of frequencies used in the inversion ($n_f$), since it is defined as (Wathelet et al. 2004):

$$misfit = \sqrt{\sum_{i=1}^{n_f} \frac{(x_{di} - x_{mi})^2}{n_f}}$$

Where $x_{di}$ and $x_{mi}$ are the velocities of the data curve and of the modelled curve, respectively.

The final misfits for the left and right side of the section are 0.02416 and 0.03797, respectively. Although the discretization of the two curves is the same (0.4 Hz), we think it is difficult to get any useful information from the comparison of these misfit values since the dispersion curves to be inverted had a very different frequency range, thus $n_f$ is not comparable. Nonetheless, we will add this information in the updated manuscript.

For the SRT inversion in pyGIMLi, we employed an initial gradient model with P-wave velocities (Vp) increasing gradually from 500 m/s at the surface to 5000 m/s at the bottom of the model domain. This range was selected based on plausible subsurface conditions for the study area. We also tested alternative initial models with both narrower and broader velocity ranges, and observed that the final inversion results remained essentially

unchanged. This suggests that the inversion is relatively insensitive to the choice of starting model. For the ERT inversion performed in ResIPy, a homogeneous initial model with a resistivity value of 10 kΩ·m was selected, as this was considered a representative value for the highly resistive environment of a rock glacier. Similar to the SRT case, we tested initial models with both higher and lower resistivity values. The resulting inverted resistivity models showed negligible differences, indicating that the inversion outcome is largely insensitive to the choice of starting model within a geologically reasonable range. In both cases, variations in the initial model may lead to slight differences in the number of iterations required for convergence, but they do not affect the final inverted model.

Due to some redundancies in the text, the authors could also consider merging the interpretation of the results in sections 4.1, 4.3 and 6.1 to provide a qualitative-only description and comparison of the ERT/SRT and MASW results in 4.1 and 4.3 and the joint interpretation in terms of subsurface materials in 6.1.

Redundancy is partially intentional. We organized the manuscript in such a way to guide the reader through a data interpretation process: the acquisition and preliminary interpretation of more conventional geophysical data (ERT and SRT), the observation of an inconsistency between them, the additional information brought by MASW, the hypothesis of the presence of a supra-permafrost water saturated layer and the synthetic modelling. We believe that maintaining the current structure would facilitate a better understanding of the different steps in our work.

A minor suggestion to improve the comparability of the geophysical results would also be to superimpose the final S-wave velocity models from MASW onto either Figure 2 or Figure 7. Alternatively, an additional figure comparing the final results across methods could be included instead. I think this would aid the reader in identifying and comparing the (structural) similarities between the three methods.

We acknowledge Reviewer3 for suggesting the possibility of overlapping the final models from MASW with Figure 2 or Figure 7. However, we think that the present figures and the description provided in section 6.1 already give a thorough explanation of how we built our conceptual model based on the joint analysis of all geophysical methods, including MASW. Moreover, Figure 2 and Figure 7 include topography. Thus, the Vs profiles should be rotated with respect to the average topography, resulting in a poorly-attractive figure.

All in all I enjoyed reading this manuscript and believe it presents an original and relevant approach to permafrost investigations in a mountainous setting.

**References**

Wathelet M., Jongmans, D. and Ohrnberger, M., Surface-wave inversion using a direct search algorithm and its application to ambient vibration measurements, Near Surface Geophysics, 211-221, 2004.

**Supplementary Material**

**ERT Synthetic (Forward) Modeling**

Forward modeling in Electrical Resistivity Tomography (ERT) involves the numerical simulation of the electrical potential distribution in the subsurface based on a known resistivity model. This process requires solving Poisson's equation, which describes the behavior of the electric field generated by current injection through electrodes placed on the surface or in boreholes (Binley & Slater, 2020). Forward modeling is a crucial step in the ERT workflow, as it allows for the prediction of the theoretical response of the subsurface for a given resistivity distribution and electrode configuration. It is commonly used to test the effectiveness of specific electrode arrays, assess the system's sensitivity to subsurface resistivity variations, and validate the quality of inversion results (Loke et al., 2003; Binley & Kemna, 2005).

In this study, forward modeling of ERT was performed using the open-source software ResIPy (Blanchy et al., 2020). The objective was to evaluate whether the electrode array and acquisition configuration used during the measurement campaign at the Flüelapass rock glacier provided sufficient resolution to detect a thin layer of water-saturated sediment overlying the permafrost. We hypothesize that this layer may have contributed to the attenuation of P-wave propagation at depth.

The forward model was based on the subsurface structure shown in Figure 4 of the manuscript, with electrical resistivity values assigned to each layer according to the inverted resistivity model derived from field data (Figure 2a of the manuscript). Specifically, resistivities of 20 kΩ·m, 10 kΩ·m, 5 kΩ·m, and 100 kΩ·m were assigned to the surface debris layer, compact sediment layer, bedrock, and frozen layer, respectively (Figure S1). A representative value of 1 kΩ·m was assigned to the water-saturated sediment layer. In rock glacier environments, such layers can exhibit resistivities on the order of 1000 Ω·m, depending on factors such as material composition, water chemistry, and temperature. This value is plausible particularly when the substrate consists of coarse, blocky debris with large pore spaces and low clay content, which reduces electrical conductivity even under saturated conditions (Hauck & Vonder Mühll, 2003; Hilbich et al., 2021). Additionally, if the pore water has low ionic content—as is typical of glacial meltwater—the resulting electrical conductivity remains low, yielding higher resistivity values (Hauck, 2002). Cold yet unfrozen conditions, or partially saturated porous media, may also lead to resistivities within this range.

[Figure]

*Figure S1: Conceptual model used for the synthetic ERT modelling.*

The synthetic dataset was generated using a dipole–dipole multi-skip acquisition scheme identical to that employed in the field survey, with an array of 48 electrodes spaced 3 meters apart. A 5% noise level was added to the synthetic measurements, consistent with the estimated noise in the real dataset. The synthetic data were then inverted using the same parameters applied to the inversion of the real dataset, resulting in the resistivity model shown in Figure S2. The color scale used corresponds to that of the electrical resistivity model obtained from the real data, presented in Figure 2a of the manuscript.

[Figure]

*Figure S2: Synthetic electrical resistivity model derived from forward modeling applied to the conceptual model presented in Figure S1.*

As shown in Figure S2), the result does not clearly reveal the presence of the thin water-saturated sediment layer overlying the frozen layer, confirming that the ERT survey conducted at the Flüelapass rock glacier site lacked the resolution and configuration necessary to resolve such a feature. This limitation is likely due to the relatively large electrode spacing.

Compared to the real electrical resistivity model (Figure 2a of the manuscript), slight deviations can be observed, which can be attributed to the simplifications adopted in the conceptual model (Figure 4 of the manuscript and Figure S1). The conceptual model used for the synthetic simulation does not account for the natural heterogeneity typically encountered in the field, including lateral and vertical variations in layer thickness, composition, and continuity. As in the seismic forward modeling, we assumed laterally homogeneous, planar layers and excluded surface topography, resulting in an idealized representation intended to enhance the theoretical detectability of the target layer.

**Ray density in Seismic Refraction Tomography (SRT)**

Figure S3 shows the Vp model obtained through Seismic Refraction Tomography (SRT) together with the computed ray paths. In the first half of the model domain (0 < x < 60 m, where we assume the absence of the thin water-saturated layer), ray coverage is well distributed both at the shallow and intermediate depths. Conversely, in the second half of the section (x > 60 m, where we hypothesize the presence of the saturated layer above the permafrost), the majority of rays are concentrated within the near-surface portion of the model, with only a limited number of rays penetrating to deeper levels.

[Figure]

*Figure S3: Vp model obtained through seismic refraction tomography (SRT), together with the computed ray paths.*

**References**

Blanchy, G., Loke, M. H., Ogilvy, R., & Meldrum, P. (2020). ResIPy: A Python-based GUI for 2D/3D resistivity modeling and inversion. Journal of Open Source Software, 5(54), 2432. https://doi.org/10.21105/joss.02432.

Binley, A., & Kemna, A. (2005). DC resistivity and induced polarization methods. In Rubin, Y. & Hubbard, S.S. (Eds.), Hydrogeophysics (pp. 129–156). Springer. DOI: 10.1007/1-4020-3102-5.

Binley, A., & Slater, L. (2020). Resistivity and Induced Polarization: Theory and Practice. Cambridge University Press. DOI: 10.1017/9781108685955.

Hauck, C. (2002). Frozen ground monitoring using DC resistivity tomography. Geophysical Research Letters, 29(21), 2016. https://doi.org/10.1029/2002GL014995.

Hauck, C., & Vonder Mühll, D. (2003). Detecting seasonal changes in permafrost using geophysical methods. Permafrost and Periglacial Processes, 14(3), 213–222. https://doi.org/10.1002/ppp.451.

Hilbich, C., Fuss, C., Mollaret, C., Hauck, C., & Hoelzle, M. (2021). Multi-decadal geophysical monitoring of permafrost evolution in mountain terrain – The PACE legacy. The Cryosphere, 15(11), 5121–5145. https://doi.org/10.5194/tc-15-5121-2021.

Loke, M. H., Acworth, I., & Dahlin, T. (2003). A comparison of smooth and blocky inversion methods in 2D electrical imaging surveys. Exploration Geophysics, 34(3), 182–187. DOI: 10.1071/EG03182.